# Understanding Target-Specific Effects of Antidepressant Drug Pollution on Molluscs: A Systematic Review Report

**DOI:** 10.3390/toxics13121043

**Published:** 2025-12-02

**Authors:** Maurice E. Imiuwa, Alice Baynes, Edwin J. Routledge

**Affiliations:** 1Environmental Sciences, College of Health, Medicine and Life Sciences, Brunel University London, Uxbridge UB8 3PH, UK; alice.baynes@brunel.ac.uk; 2Centre for Pollution Research and Policy, Brunel University London, Uxbridge UB8 3PH, UK

**Keywords:** emerging contaminants, pharmaceutical pollution, non-target organisms, hazard characterization, wildlife conservation

## Abstract

Antidepressant drugs (ADDs) are one of the most widely prescribed pharmaceuticals globally and are widely detected in the environment. They are designed to target monoamine neurotransmission—a highly conserved pathway between humans and animal species. Monoamines are particularly involved in the regulation of a wide array of key biological functions in molluscs, an ecologically important group of animals. Despite this, the target-specific effects of environmental concentrations of different classes of ADDs in molluscs remain poorly understood. The present study seeks to understand the target-specific effects of environmental concentrations of different classes of ADDs in molluscs through a systematic review of the literature. This study, following our published systematic review protocol, analyzed 51 studies after deduplication and screening of 1156 identified records. Included studies reported on a range of outcomes, including reproductive, (loco)motor, developmental, behavioral, immuno-modulating and neurophysiological effects. Data synthesis was performed with a harvest plot of exposures, effect direction and risk of bias. We found evidence (weak to moderate) for (i) immunosuppression, resulting from tissue serotonin level elevation, (ii) larval development impairment, and (iii) acetylcholinesterase inhibition, at environmental concentrations of ADDs. Most study outcomes, in addition to data-deficient outcomes, were inconclusive due largely to methodological limitations such as the use of wild-derived species with minimal or no acclimation (65.38% of included studies), lack of proper controls and replicates, and unrealistic exposures, affecting the reliability of existing data. Although the observed effects—particularly immunosuppressive ones—provide critical insight into the ecological risks posed by ADDs, their impacts at environmentally relevant concentrations remain poorly characterized for most endpoints. Given the ecological importance of molluscs, further studies addressing the identified methodological and research gaps are urgently needed to better characterize the hazards posed by environmental concentrations of ADDs.

## 1. Introduction

The widespread occurrence of human pharmaceuticals in the environment is one of the most challenging threats to ecological health [1,2]. A wide variety of pharmaceuticals, with different mechanisms of action, are routinely detected in the aquatic environment. Therefore, potential impacts on the environment by these compounds are likely to be complex and diverse, and strongly influenced by chronic exposures. And effects of chronic exposures of biological systems urgently require characterization [3,4]. Of particular concern are drug target-specific effects, which may occur when certain pharmaceuticals act on biomolecules (that are highly conserved between humans and aquatic species) at concentrations reported/measured in the environment [5,6]. Antidepressant drugs (ADDs) are a prominent example, as they target monoamine neurotransmission pathways that are widely conserved across animal phyla, including invertebrates [7,8,9,10], and exert pharmacological effects at low therapeutic concentrations [11]. As one of the most widely prescribed pharmaceutical groups globally, ADDs are also widely detected in the aquatic environment at ng/L to µg/L levels [12,13,14,15,16].

Monoamines (including serotonin, norepinephrine and dopamine) are key neurotransmitters characterized by a single amine group [17]. Although monoaminergic neurons represent a small fraction of vertebrate central nervous system, they project broadly and modulate a wide range of brain functions including emotion, behavior and cognition [18,19,20], and have additional regulatory roles across multiple organ systems. Their physiological actions are coordinated by synaptic reuptake transporters, receptors, and deaminating enzymes, which together determine the intensity and duration of monoamine signaling [21,22,23]. These biomolecules are the pharmacological targets of ADDs, which potentiate monoamine activity [24].

ADDs are widely used psychotropic drugs for depression and other mental disorders, including obsessive–compulsive disorder, panic disorder, generalized and social anxiety disorders, and specific phobias [25,26,27]. Their global consumption continues to rise in parallel with the increasing prevalence of depression [12,28,29]. The four major classes of ADDs (monoamine oxidase inhibitors (MAOIs), tricyclic antidepressants (TCAs), selective serotonin reuptake inhibitors (SSRIs), and serotonin norepinephrine reuptake inhibitors (SNRIs)) [26,30] are defined by their mechanisms of action and chemical structures, with a fifth heterogeneous group classified as atypical or ‘other’ antidepressants [31,32]. Briefly, MAOIs inhibit the monoamine-metabolizing enzymes; TCAs inhibit serotonin and norepinephrine transporters while also interacting with several post-synaptic receptors; SNRIs similarly inhibit serotonin and norepinephrine transporters but with minimal receptor interactions; SSRIs selectively inhibit the serotonin transporter; and atypical ADDs exert diverse receptor- and transporter-mediated actions [31,32]. Although ADDs are readily metabolized, excreted residues contain pharmacologically active parent compounds and metabolites [33,34,35,36], many of which pass through wastewater treatment processes only partially removed [37,38]. As a result, ADDs enter the environment via treated effluents [39,40,41], sewage sludge and reclaimed water applied to land [42], and are widely detected in the aquatic environment and wildlife [43,44,45].

While surface water concentrations of ADDs predominantly fall within the low ng/L to µg/L range, they are designed to modulate a highly conserved neurotransmission pathway at low therapeutic plasma levels. Consequently, numerous field and laboratory studies have investigated the potential ecological effects of ADD exposure in aquatic organisms. However, target-specific effects of different classes of ADDs at concentrations measured in the environment remain poorly understood [16,46,47]. Importantly, data suggest that molluscs may be particularly vulnerable due to the extensive role of monoamines in the regulation of molluscan physiology. In many molluscan species, monoaminergic neurons, including large serotonergic neurons innervating the muscular foot, or other major muscles, modulate essential processes such as feeding, locomotion, and reproductive behaviors [48,49,50,51,52,53,54]. A plausible direct role of serotonin in reproduction is supported by evidence of serotonin receptor expression on oocyte membranes [55,56], and fluctuations in gonadal monoamine levels across reproductive cycles [57,58,59]. In contrast, key vertebrate reproductive steroid receptors are absent or non-functional in molluscs. For example, steroid estrogens do not bind to molluscan estrogen receptor analogs [60,61,62], and androgen receptors have not been identified, even with extensive genomic analyses [61,63,64,65]. Interestingly, dopamine and serotonin have also been shown to regulate additional processes of reproductive development, including perivitelline fluid secretion in gastropods [66,67], serotonin-mediated cAMP-regulated larval development [68] and shell formation in the Pacific oyster [69,70].

Molluscs constitute a highly diverse and ecologically important group of animals, second only to arthropods in terms of number of species [71]. Their wide range of microhabitat adaptations, ecosystem service, and significance as a human food source through fisheries and aquaculture underscore their relevance [72,73]. Their comparatively low motility, with many species being sessile or sedentary [74], also makes them valuable indicators of chemical pollution. Monoamines in molluscs are produced in the nervous system, where they mediate chemical communication or exert hormonal actions when released into the hemolymph [75,76,77,78]. Neurosecretory cells (hormone-producing neurons), together with their targets, also contribute to the neuroendocrine system, the primary source of hormones in molluscs [79,80]. Importantly, molluscs possess the pharmacological targets of ADDs, including reuptake transporters, receptors, and oxidases [10,81,82,83,84,85], and their effective ADD concentrations overlap with levels that are reported in the environment [86,87,88]. Existing reviews on ADD effects in molluscs are narrative, and largely embedded in broader assessments of antidepressant impacts across aquatic organisms, with limited detail on molluscs specifically [47,89,90,91,92,93]. Given the growing body of literature in the field, the present study aims to synthesize the target-specific effects of different classes of ADDs in molluscs at environmental concentrations through a systematic review of the literature to inform regulatory risk assessment and guide future research.

### Objectives of the Review

The primary research question of the present study is ‘What are the target-specific effects of environmental concentrations of different classes of antidepressant drugs on aquatic molluscs?’. Accordingly, the study PECO (Population, Exposure, Comparator and Outcomes) framework was defined as follows:Populations—all species of aquatic molluscs;Exposure—laboratory-based water-born antidepressants (or their major pharmacologically active metabolites) singly administered;Comparator—naive (or vehicle solvent) control;Outcomes—physiological processes regulated by monoamines in molluscs, namely intercellular signaling events, neurophysiology [76,78], feeding [94,95], respiration [96,97], locomotion [98,99], behavior [100,101], reproduction [59,67], development [68,102] and immunity [103].

## 2. Methods

The systematic review protocol used for this study, which also included a PRISMA-P checklist, was developed in line with the guidelines and standards for evidence syntheses in environmental management by the Collaboration for Environmental Evidence (CEE) and is published [104].

### 2.1. Deviation from the Systematic Review Protocol

The following deviations from the protocol were procedurally made upon commencing the review. Firstly, the open access online tool supporting the conduct of evidence syntheses, Cadima [105], and not EPPI-Reviewer, originally described in the protocol, was used for managing articles across all stages of the review because of its adaptability [106]. Secondly, to better validate exposure, one of the 10 signaling questions used for the critical appraisal of all included studies which read ‘Are the test concentrations environmentally relevant or were the internal (tissue) levels determined?’ was modified to ‘Are the test concentrations environmentally relevant?’ while the one which read ‘Were the exposure concentrations experimentally determined in the exposure medium?’ was modified to ‘Were the exposure concentrations experimentally determined in the exposure medium or were internal (tissue) levels determined?’. Furthermore, as a surprisingly large number of studies used molluscs collected from the wild with little or no acclimation, the study quality classification (risk of bias) initially defined, using 10 signaling questions, as low (≥8), medium (6–7) and high (≤5) was adjusted to low (≥9), medium (6–8) and high (≤5), respectively, affecting only the low and medium categories. Interestingly, with this approach, only six (11.8%) of all included studies were affected [107,108,109,110,111,112], and were reclassified as ‘medium risk of bias’ from ‘low risk of bias’. Finally, we now, more appropriately, refer to the ‘medium risk of bias’ as ‘some concerns’.

### 2.2. Search Strategy, Article Screening and Critical Appraisal

Article search, screening and critical appraisal of included studies proceeded as detailed in the systematic review protocol [104]. Briefly, both bibliographic database and gray literature searches were conducted in the present study. The bibliographic databases included Web of science, Scopus, and PubMed using search strings developed for database-specific retrieval sensitivity and relevance, and this search was managed on Cadima. The supplementary gray literature sources included ProQuest Dissertations and Theses Global, Open Access Theses and Dissertations, OpenGrey, Grey Literature Report, Research square and EcoEvoRxiv, using key study terms. Following an initial search (bibliographic databases only) conducted in November 2023, an updated search (bibliographic databases and gray literature) was conducted in March 2024. Furthermore, article screening (title and abstract, and then full-text) was carried out using the predefined inclusion and exclusion criteria based on the study PECO framework.

Finally, critical appraisal of all included studies was carried out using a set of 10 predefined signaling questions (described in Results section below) across various study selection (baseline), performance and reporting domains. Depending on the sum of the answer to the signaling questions (yes = 1, no = 0), each of the included studies was classified as either low risk of bias (≥9), some concerns (6–8), or high risk of bias (≤5). Article search strings were developed in consultation with an academic liaison librarian, while article search, screening and critical appraisal were carried out by one reviewer and were independently verified by two further reviewers for completeness and consistency.

### 2.3. Data Extraction, Potential Effect Modifiers and Synthesis

Data extraction and synthesis were carried out as detailed in the systematic review protocol [104]. The extraction was performed by one reviewer on Cadima to facilitate data organization and was evaluated by two reviewers for completeness and consistency. Briefly, all data on the study PECO framework and other study characteristics including test organisms (class, species, life stage, habitat type and source), test antidepressants (type and class), control, exposure (type, concentrations, duration, and medium renewal frequency/level), antidepressant concentration measurement (exposure medium, hemolymph and tissue burden), number of replicates, endpoint studied, effects (including effect concentrations) and statistical analyses were extracted.

In line with the systematic review research question, environmentally relevant concentrations for the various antidepressants reported in all studies included in this review were also defined. A total of 1 µg/L was defined as the upper limit of environmentally relevant concentrations for fluoxetine (SSRI) as levels in the region of 1 µg/L has been reported in treated effluents [113,114] and up to 330 ng/L in surface water [114], while also accounting for typical fluctuations in exposure concentrations between medium renewals in laboratory studies. Importantly, the 1 µg/L fluoxetine threshold was applied only when exposure concentrations were analytically measured, as actual concentrations can be higher than the nominal [115].

For antidepressants with higher surface water levels, the maximum reported levels were used to define the upper limits of environmentally relevant concentrations. These include 3.35 µg/L for amitriptyline (TCA) [116], 55 µg/L for venlafaxine (SNRI) [117] and 76 µg/L for citalopram (another SSRI) [118]. For atypical antidepressants and other members of SSRIs, SNRIs and TCAs with quite variable surface water levels, the maximum reported surface water concentrations were likewise used [16]. Although these maxima may represent relatively rare or high-end environmental scenarios, their inclusion ensures a comprehensive characterization of the full range of exposure levels documented in monitoring studies. Furthermore, potential effect modifiers, i.e., factors with potential to cause some degree of heterogeneity in the observed effects of antidepressants in molluscs were defined in the study protocol around characteristics of the test organisms, type of antidepressants and the nature of exposure.

General study characteristics of all included studies were synthesized and presented using summary figures while a narrative synthesis of study primary data was completed using a harvest plot as the data are not amenable to meta-analyses on account of study heterogeneity [119]. Note that exposure effects described here (an observed increase or decrease in a variable under investigation) are those that are significantly different from their respective controls (comparators). As a result, the term ‘significantly different’ is not used throughout the discussion except for clarification in outcomes with apparent exposure-induced effects for trends that are not significantly different from the control.

### 2.4. Evidence Synthesis

Evidence synthesis was performed by integrating each study outcome and the study risk of bias with effect concentrations (environmentally relevant or higher) in line with the systematic review research question. This resulted in four (4) levels of evidence, including (i) ‘very little or no evidence’ for effects that occurred at environmentally relevant concentrations in studies with high risk of bias, (ii) ‘weak to moderate evidence’ for effects that occurred at environmentally relevant concentrations in studies with ‘some concerns’ risk of bias, (iii) ‘strong evidence’ for effects that occurred at environmentally relevant concentrations in studies with low risk of bias, and (iv), ‘generally or definitely inconclusive’ for effects that occurred at concentrations that are not environmentally relevant in studies across all 3 categories of study risk of bias. Adapting the EPPI-center approach [120], the overall weight of the synthesized evidence was further evaluated based on the evidence level that has the highest number across all studies for each of the study outcomes.

## 3. Results

### 3.1. Study Selection

The result of our study selection process is shown in Figure 1. Briefly, the bibliographic database search identified 1156 records while none (0 records) were identified from the supplementary gray literature search. Following duplicate removal (295 records), 861 records were screened at the title and abstract level, 793 were excluded as ineligible records while the remaining 68 records were included for full-text screening. After full-text screening, 51 studies were selected and included in the systematic review. The list of included studies (with Cadima-assigned article ID) is provided as an additional data file (Appendix A), while that of excluded studies, with reasons, is provided in the Appendix A.

### 3.2. Characteristics of Included Studies

General study characteristics of all included studies are presented in Figure 2, Figure 3 and Figure 4. Briefly, SSRIs were the most reported antidepressants (76.40%), followed by SNRIs (13.48%), TCAs (7.87%) and atypical antidepressants (2.25%), from 1998 to January 2024, while there were no reports on MAOIs. Three classes of molluscan species were reported, that included bivalves (52.94%), gastropods (36.77%) and cephalopods (10.29%). With one of the studies using both wild- and aquaculture-derived species, a total of 34 (65.38%) of the included studies used molluscs derived from the wild, while 9 (17.31%), 8 (15.39%) and 1 (1.92%) used molluscs derived from aquaculture, laboratory culture and mesocosm, respectively. The time period from collection to exposure for wild- and aquaculture-derived molluscs were often rather short. These ranged from 1 to 3 days (26.83%), from collection to exposure (without information on acclimation period), to several months (2.44%), from collection to exposure (including acclimation period). Standard regulatory tests with molluscs (i.e., published by the Organisation for Economic Co-operation and Development (OECD)) primarily recommend using animals which have been bred and routinely cultured under laboratory conditions. When wild-derived molluscs are needed, the OECD recommends the use of F1 offspring raised under controlled laboratory conditions, rather than wild-collected F0 animals [121,122]. Exposure type and duration in the included studies were also markedly different, and ranged from static (1–4 h: 25.00%) to flowthrough (42–44 days: 5.77%).

A broad range of outcomes were reported; reproduction and locomotion endpoints were the most frequent. The others are cell signaling, immunity, neurophysiology, predatory and cryptic behaviors, general behaviors, development and respiration. It is noteworthy that only 20 (39.22%) of the included studies analytically determined exposure concentrations in the exposure medium, while 6 (11.76%) reported tissue levels of ADDs. Plasma (hemolymph) levels, which are particularly key to understanding the potential of human pharmaceuticals to exert pharmacological effects in non-target organisms with conserved drug targets [123], were not determined in any of the included studies. Furthermore, the following quantitation methods were used for the measurements of hemolymph/tissue levels of various physiological parameters, including coulter counter electronic particle counter/size analyzer and Neubauer hemocytometer (hemocyte count/proliferation) [124,125]; enzyme immunoassay (EIA) (hemolymph γ-aminobutyric acid (GABA), acetylcholine (Ach) and 3’,5’-cyclic adenosine monophosphate (cAMP) levels) [108,125,126,127]; PepTag non-radioactive PKA assay (protein kinase A activity) [108,126,127]; mRNA quantification (qPCR) and fluorescent substrate rhodamine B procedure (levels of ATP-binding cassette (ABC) transporters) [87,108,112,126,127]; spectrophotometric procedure (acetylcholinesterase (AChE) activity) [124,126,128,129]; tritiated antagonist ([3H]-MK801) binding assay (N-methyl-D-aspartate-like receptor binding) [130]; and mRNA quantification (qPCR) and high-performance liquid chromatography (HPLC) system (monoamine levels) [108,115,126,127,130,131,132]. Additionally, as there is a lot of uncertainty regarding the endogenous synthesis of vertebrate-type steroids in molluscs and the suitability of immunoassays for their quantitation [61,133], cortisol level data in one of the included studies [125] was not reported in the present study. Extracted data for all included studies are provided as an additional data file (S3 File).

### 3.3. Critical Appraisal

The results of the critical appraisal (internal validity) of each of the included studies (51) using a set of 10 signaling questions across various domains (baseline, performance and reporting) of laboratory exposure studies, and the overall risk of bias, are presented in Figure 5. Of the 51 studies, 8 (15.69%) had low risk of bias, 33 (64.70%) were assessed to have some concerns, while 10 (19.61%) were assessed to have high risk of bias. The ‘some concerns’ study risk of bias category was, amongst other quality parameters, majorly affected by the source of the test organisms and acclimation period, while the ‘high risk of bias’ study category was, in addition to these parameters, considerably affected by other quality parameters such as experimental control, exposure medium renewal, replicates and exposure validation (analytical determination of test chemicals). Furthermore, it is noteworthy that of the 42 studies (82.35%) assessed to have used environmentally relevant concentrations (Figure 5a), 11 (21.57%) only used a single exposure concentration level, 9 (17.65%) with multiple exposure concentration levels only had a single concentration that is environmentally relevant, while 5 additional studies (9.80%) with multiple bioassays only had environmentally relevant concentrations in one of the assays. Critical appraisal data is provided as an additional data file (S4 File).

### 3.4. Reported Outcomes and the Range of Evidence Across All Studies for Each Outcome

The result of the synthesis using a harvest plot is presented in Figure 6. The synthesis, which integrated all outcomes for each of the included studies and the study risk of bias with effect concentrations of ADDs (environmentally relevant or higher), reveals eight (8) categories of effects depending on the levels of evidence (Table 1). The synthesized evidence is further evaluated (overall weight) based on the evidence level that has the highest number across all studies for each outcome (Figure 6).

## 4. Discussion

### 4.1. Effects of ADDs on Monoamine Neurotransmission

In order to gain insight into the potential mechanisms of action of antidepressant drugs (ADDs) in molluscs, their effects on monoamine neurotransmission (the primary pharmacological targets) are examined first, before the more overt apical response outcomes. The up-regulation of serotonin transcription (mRNA) levels in hemocytes (Cadima ID: 163) and tissues (IDs: 337 and 610) following exposure to fluoxetine (0.03–300 ng/L, nominal), and the increase in tissue serotonin and epinephrine levels (ID: 271) following exposure to venlafaxine (10 µg/L, measured) in *M. galloprovincialis* are consistent with the known mechanisms of action of these ADDs. These drugs bind to and inhibit monoamine reuptake transporters, thereby increasing synaptic or tissue concentrations of monoamines.

The observed decrease in serotonin levels (day 1), and absence of effects on epinephrine levels in female *M. galloprovincialis* (ID: 271) may reflect sex-related differences, as both monoamines were elevated in males, and serotonin was already rising in females by day 7 when the exposure ended. Sex differences in response to ADD treatment are also documented in humans, with women often responding more slowly to some ADDs [167].

Brain monoamine levels in *S. officinalis* are generally low and vary across brain regions [168,169]. Therefore, the use of whole-brain samples rather than region-specific samples may account for no observed effect on brain monoamine levels reported in *S. officinalis* ADD exposure studies (IDs: 41, 495 and 508). These include a study which measured serotonin, norepinephrine and dopamine levels following fluoxetine exposure (1 and 100 ng/L, nominal; ID: 495); a second study measuring serotonin and dopamine levels following fluoxetine exposure (1.6 and 18 µg/L, measured; ID: 41), in which fluoxetine was detected in the control which may have compromised the study; and a third study where serotonin, norepinephrine and dopamine levels were measured (evaluated on days 10 and 20) following venlafaxine exposure (5 and 100 ng/L, measured; ID: 508), where only a small, but statistically significant, reduction in norepinephrine level was observed by day 20.

The observed increase in tissue levels of serotonin by ADDs (weak to moderate evidence) has the potential to interfere with cellular processes regulated by serotonin, depending on the tissue serotonin receptor-subtype that is activated. The decrease in cyclic adenosine monophosphate (cAMP) and protein kinase A (PKA) (IDs: 163, 337, 610) in *M. galloprovincialis* (weak to moderate evidence) is consistent with the activation of G protein-coupled serotonin receptors. Serotonin receptors (5-HT_1/5_) coupled to G-proteins containing α_i/o_ subunits inhibit adenylate cyclase (AC) upon activation, resulting in a decrease in intracellular levels of cAMP and cAMP-dependent PKA [9,170].

Adenylate cyclase (AC) catalyzes the conversion of ATP to cyclic adenosine monophosphate (cAMP), a ubiquitous intracellular second messenger with a key role in signal transduction using various effectors, including protein kinase A (PKA) [171]. PKA activates target proteins by phosphorylation, a mechanism that is involved in nearly all cellular regulatory processes. Inhibition of cAMP/PKA pathway appears to be involved in a wide range of downstream intracellular effects [172]. For example, inhibition of ATP-binding cassette (ABC) transporters observed in the mussels, *M. galloprovincialis* (IDs: 163, 337 and 610) and *D. polymorpha* (ID: 294), and in the clam, *C. fluminea* (ID: 544), following exposure to fluoxetine and citalopram mirrors the inhibitory action of ADDs on human P-glycoprotein [173]. These transporters have been shown to be transcriptionally regulated by the cAMP/PKA pathway in molluscs [108]. Likewise, the increase in lysosome/cytoplasm volume ratio observed in *M. galloprovincialis* (ID: 36), an indicator of inhibited lysosome catabolic degradation product removal, or exocytosis [174], has been demonstrated to be inducible by cAMP pathway inhibition [175,176]. Furthermore, the reduced hemocyte viability, proliferation and phagocytotic activity in *T. granosa* (ID: 37) may reflect impaired ATP-dependent PKA activity, which is central to molluscan immune defense [177].

These studies demonstrate that ADDs inhibit the xenobiotic efflux activity of ABC transporters (multixenobiotic resistance (MXR) system) in molluscs (IDs: 163, 294, 337, 544 and 610), a notable contrast to the induction of MXR typically caused by other aquatic contaminants [178]. Collectively, these observations support the conclusion that environmental levels of ADDs (fluoxetine and citalopram) exert immunosuppressive effects (Figure 7) potentially increasing vulnerability to co-occurring environmental stressors.

### 4.2. Effects of ADDs on Other Neurotransmitter Pathways

The effects observed on components of non-monoaminergic neurotransmission pathways, including acetylcholinesterase (AChE) activity, acetylcholine (ACh) and GABA levels, and glutamate NMDA-like receptor binding are generally consistent with the mechanism of action of ADDs on these targets in humans [179,180,181]. Reduced AChE activity was observed in various tissues in *V. philippinarum* (ID: 6), *P. perna* (ID: 250) and *M. galloprovincialis* (IDs: 337 and 561) exposed to various concentrations of fluoxetine (all nominal except ID: 250). Increase in ACh levels in *T. granosa* (ID: 37) exposed to sertraline also reflect the inhibitory effect of ADDs on AChE [179]. As it is typical of enzyme inhibitors, the absence of inhibitory effect in *P. corneus* (ID: 285) exposed to citalopram and venlafaxine may have been caused by weak inhibition potency of these ADDs [182]. The reported increase in AChE activity in *M. galloprovincialis* (ID: 561) on day 3, followed by inhibition on day 15, also fits typical temporal patterns of enzyme inhibition.

Increased hemolymph GABA levels in *T. granosa* (ID: 37) are also consistent with the effect of ADDs on GABAergic system in humans [180]. This modulation is thought to result from increased serotonergic signaling on GABAergic neurons [180,183]. Reduced glutamate NMDA-like receptor binding in *Sepia officinalis* exposed to venlafaxine (5 and 100 ng/L, measured; ID: 508) aligns with the inhibitory effects of ADDs on glutamate levels in humans, likely mediated through glutamate receptor modulation [181]. However, in this study, an initial (day 10) increase in glutamate NMDA-like receptor binding occurred in the 100 ng/L venlafaxine group before a downward trend was observed on day 20 (*p* = 0.062), suggesting a temporally dynamic response.

Given the observed widespread modulation of non-monoaminergic neurotransmission by ADDs in molluscs combined with their effects on monoamines, it is conceivable that apical responses may arise from this combined disruption. Non-monoaminergic neurotransmission also regulates critical physiological processes in molluscs [184,185,186], increasing the likelihood that multiple pathways interact to shape downstream responses.

### 4.3. Effects of ADDs on (Loco)Motor Activities

Pedal sole adhesion is critical to locomotion in many molluscan species and is reported to be disrupted (sole detachment) by various ADDs. However, the concentrations of the different ADDs that induced sole detachment are higher than their environmental levels, and in most cases, by several orders of magnitude. To illustrate, increased sole detachment was observed in *G. umbilicalis* exposed to 1 mg/L fluoxetine, but not 1 ng and 1 µg/L, and in both *G. umbilicalis* and *L. stagnalis* exposed to 1 mg/L, but not 10 µg/L fluoxetine. Extended exposure of both species for 4 h at 1 ng–10 μg/L also failed to induce sole detachment (ID: 463; nominal concentrations).

In a second acute exposure study (ID: 500; nominal), sole detachment was similarly observed only at high concentrations of several ADDs across five molluscan species:*L. americanum* exposed to fluvoxamine (induced at 43.4 and 434 µg/L, and 4.34 mg/L, but not 4.34 ng/L and 21.7 µg/L), fluoxetine (3.45 mg/L, but not 34.5 and 345 µg/L), citalopram (405 µg/L, but not 4.05 and 40.5 µg/L) and venlafaxine (313 µg/L, but not 3.13 and 31.3 µg/L);*C. funebralis* exposed to fluvoxamine (induced at 217 and 434 µg/L, and 4.3 mg/L, but not 43.4 µg/L), fluoxetine (345 µg/L and 3.45 mg/L, but not 34.5 µg/L), citalopram (2.03 and 4.05 mg/L, but not 40.5 and 405 µg/L) and venlafaxine (157 and 313 µg/L, and 3.13 mg/L, but not 3.13 and 31.3 µg/L);*T. fasciatus* exposed to fluvoxamine (induced at 434 µg/L and 4.34 mg/L, but not 43.4 µg/L), fluoxetine (345 µg/L and 3.45 mg/L, but not 3.45 and 34.5 µg/L), citalopram (405 µg/L, but not 4.05 and 40.5 µg/L) and venlafaxine (313 µg/L, but not 3.13 and 31.3 µg/L);*U. cinerea* exposed to fluvoxamine (induced at 434 and 868 µg/L, and 2.17 and 4.34 mg/L, but not 43.4 µg/L), fluoxetine (3.45 mg/L, but not 34.5 and 345 µg/L), citalopram (not induced at all at 4.05, 40.5 and 405 µg/L, and 4.05 mg/L) and venlafaxine (3.13 mg/L, but not 3.13, 31.3 and 313 µg/L);*N. ostrina* exposed to fluvoxamine (induced at 4.34 mg/L, but not 43.4 and 434 µg/L), fluoxetine (3.45 mg/L, but not 34.5 and 345 µg/L), citalopram (4.05 mg/L, but not 40.5 and 405 µg/L) and venlafaxine (1.57 and 3.13 mg/L, but not 31.3 and 313 µg/L).

Similarly, in *P. corneus*, only very high venlafaxine concentrations (1000 and 100 µg/L, but not 1 and 10 µg/L; measured) were found to significantly induce sole detachment (ID: 285).

In sharp contrast, one study reports increased sole detachment at much lower concentrations, i.e., pg/L and ng/L rather than µg/L and mg/L:*L. carinata* (from 313 pg/L venlafaxine and 405 pg/L citalopram; nominal, ID: 7);*L. elodes* (from 31.3 ng/L venlafaxine and 4.05 µg/L citalopram; nominal, ID: 7).

It is unlikely that potential species sensitivity to ADDs alone can explain these observations which are yet to be reproduced.

In crawling or gliding gastropods, attachment to substratum is achieved by suction (pressure reduction beneath the foot by muscular contraction), clamping (force applied by the shell rim), or pedal sole adhesion (mucus-mediated flow resistance) [79,187]. Serotonin stimulates locomotion [188] and mucus secretion [99,189], with increased tenacity of attachment from low pedal sole mucus secretion [79]. Hyperactivation of pedal serotoninergic system from exposure to very high concentrations of ADDs (SSRIs and SNRIs) may therefore explain the observed sole detachment [99].

The other locomotor effects including an increase in total movement, crawling speed (SNRIs), foot protrusion, digging, mantle lure display, and a decrease in righting, crawling speed (SSRIs), algal clearance rate and neutral red filtration rate observed in different species of molluscs also occurred at concentrations of ADDs several folds and orders of magnitude higher than reported environmental levels (IDs: 20, 22, 94, 103, 219, 463, 478, 488, 495, 526, and 544).

The only exceptions to these observations were three venlafaxine studies:31.3 µg/L (nominal) venlafaxine in *I. obsoleta* (ID: 22; longer righting time);31.3 µg/L (nominal) venlafaxine in *U. cinerea* (ID: 219; increased crawling speed);31.3 µg/L (nominal) venlafaxine in *L. americanum* (ID: 219; number reaching air–water interface).

Even these cases show effect concentrations at or above environmental upper limits and require caution.

Serotonin is known to stimulate various locomotor activities in molluscs [190,191], potentially explaining the observed increase in the reported locomotor activities. TCAs also antagonize certain synaptic receptors, including serotonin receptors [192], which may underlie the longer righting times reported. Reduced crawling speed and increased righting time observed after SSRI/SNRI exposure may also reflect a tendency toward sole detachment at high ADD concentrations [99].

A decrease in algal clearance rate and neutral red filtration rate in *L. siliquoidea* (ID: 526) and *C. fluminea* (ID: 544), respectively, may reflect general toxicity, as the effect concentration in adult *L. siliquoidea* (0.3 mg/L sertraline, measured) resulted in 100% mortality in juveniles, and in *C. fluminea* the effect was only seen at the highest test concentration reported (50 µg/L fluoxetine, measured).

It is important to note that the observed effects from the majority of the locomotor outcome studies were judged ‘generally or definitely inconclusive’, due to risk of bias, high concentrations, and use of nominal doses that may exceed actual exposure levels [115], particularly those effects assumed to occur at environmentally relevant concentrations (IDs: 7, 22 and 219). Taken together, these results do not support the conclusion that locomotor effects are likely to occur at current environmental concentrations reported for individual ADDs.

### 4.4. Effects of ADDs on Egg and Egg Mass Production

SSRIs generally induce spawning and the production of egg masses in oviparous gastropods, with or without a corresponding increase in the number of eggs. These effects have been shown to be mediated by serotonin receptors in various species of molluscs [193,194,195,196]. However, the concentrations of SSRIs required to induce these responses were typically higher than their current environmental levels. Of the nine studies reporting such effects (IDs: 66, 94, 103, 356, 358, 484, 532, 540 and 574), only one study (ID: 540-zebra mussel, *D. polymorpha*) observed spawning at environmentally relevant levels of fluoxetine (20 and 200 ng/L, measured). Even in this case, spawning was inferred indirectly from the number of oocytes per follicle and spermatozoa density per tubule, whereas in three other *D. polymorpha* studies (IDs: 356, 484 and 532), spawning induction only occurred at higher concentrations of SSRIs, including fluoxetine.

In *P. acuta* chronic (44 days) exposure to fluoxetine (12, 27 and 108 µg/L measured geometric mean) increased the number of egg masses at the two lower concentrations and decreased them at the highest dose (ID: 595). Conversely, another *P. acuta* study (ID: 247), reported decreased egg masses after exposure to fluoxetine (32.27 ± 2.3 ng/L, measured) over a period of three years; the contrasting outcome to the previous study may reflect the single exposure concentration tested and the longer exposure timeframe. The absence of egg mass (and egg number) induction in *P. pomilia* exposed to fluoxetine (0.01, 1 and 100 µg/L, nominal) may be attributable to differences in reproductive sensitivity in various species of molluscs to SSRIs (ID: 107). Similar observations (no significant effect on egg numbers) were seen in chronic exposure of *V. piscinalis* to fluoxetine (1, 4.2, 13 and 69 µg/L, measured) (ID: 66).

### 4.5. Effects of ADDs on Fertilization

As neurons (and synaptic monoaminergic neuron reuptake transporters) may not be present in sperm cells or early embryos (within ~30 min post-fertilization), early developmental effects may be mediated by membrane serotonin receptors, which appear early in development and may be disrupted by ADD binding [68,197]. In *M. galloprovincialis*, pre-exposure of sperm to SSRIs (0.5–500 ng/L, nominal) reduced fertilization, whereas SNRIs including venlafaxine (having weak receptor affinity) [198,199], were mostly ineffective with the exception of O-desmethylvenlafaxine (major pharmacological metabolite of venlafaxine) but not with venlafaxine (ID: 36). Similarly, post-fertilization exposure resulted in reduced larval development by all the SSRIs in the same study, but not with any of the SNRIs (ID: 36). In another study (ID: 292), exposure to sertraline (0.01–1000 µg/L, nominal) resulted in larval disruption at 0.01 µg/L, with apparent toxicity from 100 µg/L and complete developmental failure at ≥300 µg/L. Similarly, post-fertilization exposure to high fluoxetine concentrations (0.25 and 0.5 mg/L, nominal) in *P. acuta* caused 100% mortality in *P. acuta* larvae (ID: 340). Conversely, very high nominal concentrations (up to 100 µg/L, nominal) of various classes of ADDs including SSRIs, SNRIs and TCAs were required to disrupt embryonic development in the Pacific oyster, *C. gigas* (IDs: 513 and 619). Whether this is sheer species-specific ‘hardiness’ or a methodological ‘artifact’ is unclear, as these findings were reported by only two studies, both only reporting nominal exposure concentrations.

### 4.6. Effects of ADDs on Internal Embryo Development

In viviparous (brooding) species, where embryo–larval development occurs internally rather than in the external environment, ADD effects likely represent a combination of impacts on gamete release and internal development. In the ovoviviparous gastropod, *P. antipodarum*, in which embryos develop in the brood pouch [200], chronic exposure to fluoxetine (1, 4.2, 13 and 69 µg/L, measured) increased total embryo number at low concentrations (1 and 4.2 µg/L) and decreased at 13 (albeit not significantly) and 69 µg/L (ID: 66), perhaps reflecting the initial effect of increased egg production at the low fluoxetine concentrations. However, in another *P. antipodarum* study (ID: 54), chronic exposure to fluoxetine (0.64, 2, 11, 58 and 345 µg/L, measured except the lowest concentration) decreased the total number of embryos at 11 and 58 µg/L, with no effect at 0.64 and 2 µg/L, and 100 percent mortality at 345 µg/L. Differences between the two studies may reflect sample sizes (24 vs. 60 snails, respectively, per treatment). It is important to note that there was no decrease in the number of embryos from 0.64 to 4.2 µg/L fluoxetine as observed in studies with external post-fertilization exposure.

### 4.7. Effects of ADDs on Larval Parturition

In bivalves, larval parturition (the process whereby larvae reared from fertilized eggs on gill surface are released) is mediated by serotonin receptors [153], and was also induced only at high concentrations of ADDs. In the freshwater mussels, *E. complanata, L. fasciola* and *L. cardium*, exposure to fluoxetine (0.3, 3.0, 30, 300 and 3000 µg/L; nominal) only resulted in the induction of parturition at 3000 µg/L (*L. cardium* and *L. fasciola*), while spermatozeugmata were induced in *E. complanata* at 300 µg/L and non-viable (immature) larvae were induced at 300 and 3000 µg/L (ID:103). Induction of parturition was similarly observed at high concentrations of SSRIs in other bivalve species including *S. striatinum* and *S. transversum* (ID: 356); *A. cygnea* (ID: 358); *S. striatinum* and *M. leucophaeata* (ID: 484), and *S. striatinum* (ID: 574). Interestingly, the atypical ADD, trazodone, also induced parturition in *A. cygnea* but not the potent serotonin receptor antagonist, mianserin (ID: 358). This suggests that inhibition of serotonin reuptake (and associated increases in tissue serotonin and serotonin receptor activation), rather than direct receptor antagonism, underlies the effect [201] as trazadone exhibits SSRI-pharmacology at certain concentrations as well as has weak affinity for certain serotonin receptors. Similar to the findings on locomotor effects, these studies do not support the conclusion that these reproductive effects are possible at current environmental levels for individual ADDs. Indeed, they were generally reported at concentrations several orders of magnitude higher than environmental levels of fluoxetine, and largely in studies with ‘some concerns’ regarding the risk of bias.

### 4.8. Effects of ADDs on Post-Larval Development and Neonates

Effects of ADDs on the development of post-larvae and neonates may depend on several factors including the timing and duration of exposure, and the sensitivity of the species. In *L. fasciola,* more larvae successfully completed metamorphosis on the host fish following a 24 h exposure to fluoxetine (1 and 100 µg/L, nominal), possibly reflecting accelerated cell division process caused by the timing and brevity of the exposure (ID: 94). In molluscs, serotonin has been shown to mediate metamorphosis by regulating cell division through differential expression of serotonin receptor subtypes between larval, pre-metamorphic and metamorphic stages [68,202]. While this report suggests that larval development in *L. fasciola* can be enhanced by fluoxetine, under realistic environmental conditions with chronic exposure, such effects may be reversed (i.e., developmental inhibition).

In *P. antipodarum*, a chronic 42-day exposure to fluoxetine (1, 4.2, 13 and 69 µg/L; measured) was reported to decrease in the cumulative number of neonates only at the highest test concentration, suggesting an effect on egg production (stimulation) rather than on the survival of embryos or neonates, indicating possible low neonate sensitivity to fluoxetine in this species (IDs: 66 and 67). Similar to the findings on locomotor effects, these studies on reproductive and early developmental effects do not support the conclusion that the reported effects are possible at current environmental levels of fluoxetine.

### 4.9. Effects of ADDs on Neurophysiology: Camouflage, Learning and Brain Cell Development in S. officinalis

In juveniles of the common cuttlefish, *S. officinalis*, chronic exposure to fluoxetine (5 ng/L, nominal) from hatching resulted in improved uniform body pattern camouflage but not disruptive body pattern (ID: 9). Serotonin is known to mediate the rapid retraction of chromatophores (pigment-containing organs) in cephalopods [203], thereby revealing the underlying colorless refractile leucophores that enable the skin to reflectively match a uniform background [204]. Topical application of serotonin on the skin of the squid, *Loligo opalescens*, is also known to cause paling [204,205], suggesting that the observed sensitivity of juvenile *S. officinalis* to this effect may be more of a direct local action of fluoxetine on the skin rather than a systemic effect. However, other studies reported inconsistent outcomes. In *S. officinalis* exposed from 15 days pre-hatching to 30 days post-hatching, fluoxetine (1 and 100 ng/L, nominal) caused only transient improvements to uniform body pattern camouflage at 1 ng/L (ID: 495). In another study (ID: 41), exposure of the same species (eggs to hatchlings) to fluoxetine (1.6 and 18 µg/L, measured) transiently impaired uniform body pattern camouflage at 1.6 µg/L fluoxetine. However, the study was complicated by fluoxetine detection in controls and methodological limitations (e.g., low replication, organism handling). Fluoxetine (1.6 and 18 µg/L) also decreased cell proliferation in brain regions of hatchlings associated with vision and cognition (ID: 41), likely due to the high exposure concentrations and reported contamination in the control.

In a separate learning–retention trial (‘prawn-in-the-tube’), *S. officinalis* exposed to fluoxetine (15 days pre-hatching to 30 days post-hatching at 1 and 100 ng/L, nominal) showed impaired retention (recall after 1 h at 1 and 100 ng/L) but not learning at the end of the trial (ID: 18). Probable ‘systematic’ increase in serotonin level may have resulted in an overwhelming increase in feeding stimulation (SSRIs are also known to generally enhance feeding [206]), whereas memory regulation by serotonin—observed as the loss of retention—is known to be brain region-specific in molluscs [207,208].

Taken together, these studies, with high concentrations and control contaminations, do not support the conclusion that these effects are possible at current environmental levels of fluoxetine. Additionally, the conflicting outcome data on the effects of fluoxetine on uniform background camouflage, as they stand, in *S. officinalis*, requires further studies for clarification.

## 5. Robustness of the Synthesis

While evidence synthesis is generally limited by the methodological quality of the included studies [209], including, as in our case, the number of studies and molluscan species evaluated for each outcome, there is a plethora of reports for the majority of the outcomes explored in the present study, providing sufficient data to describe effect-types as well as their magnitude and direction. The integration of individual outcomes and study risk of bias (internal validity) with effect concentrations (environmentally relevant or higher) allowed for a robust synthesis in line with the study objective for each outcome. Furthermore, by adapting the EPPI-Center approach [120], the overall weight of the synthesis for each outcome was further evaluated using the highest number of the different levels of evidence across all studies for each outcome. 

## 6. Conclusions and Future Research Direction

This is the first systematic review on the effects of antidepressant drug (ADD) pollution in molluscs, an ecologically important and taxonomically diverse group of animals. In line with the study research question, we found weak-to-moderate evidence of target-specific effects of environmental concentrations of ADDs in molluscs, based on the overall weight-of-evidence assessment across outcomes. The effects are (1) modest increases in tissue levels of serotonin in molluscs following exposure to SSRIs and SNRIs, with downstream modulation of serotonin-regulated intracellular signaling pathways, (2) acetylcholinesterase inhibition, and (3) larval development disruption, likely via receptor-mediated pathways. Of particular importance, disruption of serotonin-mediated intracellular signaling was strongly associated with immunity impairment, including inhibition of 5-HT_1_-like receptor pathways, reduced cAMP and protein kinase A activity, and consequent suppression of ABC transporter function, components central to molluscan immune defense.

Demonstrable changes in more overt apical endpoints that likely require substantially higher internal monoamine levels were typically observed only at ADD (SSRIs and SNRIs) concentrations far exceeding those currently detected in the environment. Moreover, several potentially important outcomes remain data-deficient (i.e., supported by only one study), including aspects of immunity (reduced hemocyte phagocytosis and viability; increased lysosome/cytoplasm ratio), reduced external fertilization, and neurotransmission-related effects (increased acetylcholine and hemolymph GABA levels; altered NMDA-like receptor binding). These merit further investigation, especially given that some were reported at environmentally relevant concentrations and in studies not classified as high risk of bias.

Inconclusive outcomes were also common where results were inconsistent across studies or occurred only at high, non-environmental exposure levels, particularly for locomotor (sole detachment, righting time, crawling speed, total movement, digging behavior, algal clearance/filtration and mantle lure display), reproductive (spawning, egg mass/egg production, gonadal development/gonadosomatic index, embryonic development, larval parturition and juvenile development), neurophysiological (disruption of memory and camouflage, and modulation of (nor)epinephrine levels) and decreased hemocyte count/proliferation.

Across most of the included studies, several methodological limitations were identified. These included minimal or no acclimation of wild-derived organisms, insufficient controls, short behavioral assay durations, reliance on nominal concentrations, non-replication and unrealistic dosing. Each of these reduce confidence in many published findings. Addressing these limitations will substantially strengthen future research. Additional priorities include the following: (1) environmentally realistic, analytically verified exposures (QA/QC reported); (2) greater inclusion of non-SSRI ADD classes (e.g., TCAs and atypical); (3) measurement of internal (hemolymph) ADD concentrations to link external exposure with organismal sensitivity; (4) targeted research on data-deficient outcomes; and (5) wider evaluation of feeding, an essential physiological process in molluscs.

From a broader ecological and regulatory perspective, it is crucial to determine whether the mechanistic disturbances identified (and particularly the immunosuppressive effects, which may alter host defense, infection susceptibility, and resilience to additional stressors) translate into ecologically meaningful consequences. Understanding ecological risk requires establishing whether exposed populations survive, reproduce, and maintain stable population trajectories under chronic environmental exposures. If populations persist, regulators must also consider whether ADD exposure induces selection pressures, favoring tolerant genotypes or phenotypes over successive generations. These questions fall within the domain of evolutionary toxicology and require multigenerational or transgenerational studies, which are notably lacking for ADDs in molluscs.

Finally, although this review focused on single-compound exposures, understanding individual ADD effects at environmentally realistic concentrations is essential for predicting mixture effects under both laboratory and real-world exposure scenarios. We show that environmental ADD concentrations can modulate serotonin signaling, inhibit acetylcholinesterase, and disrupt larval development, and importantly, impair immune processes. While the identified immunosuppressive effects have potential utility in shaping regulatory thoughts on how to protect wildlife from the effects of environmental antidepressants, the identified research and methodological gap is a critical and high-priority research need.

## Figures and Tables

**Figure 1 toxics-13-01043-f001:**
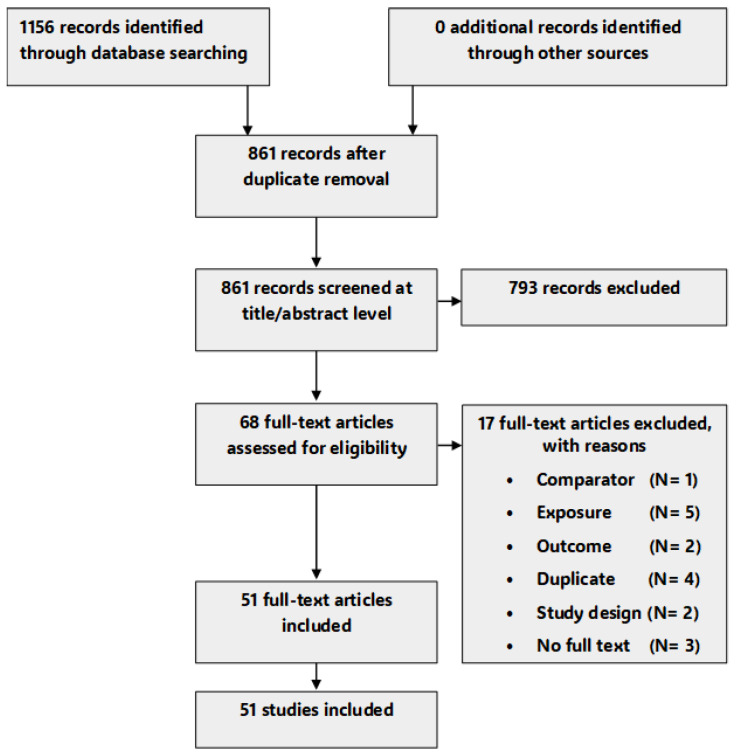
Study flow chart for article search and selection.

**Figure 2 toxics-13-01043-f002:**
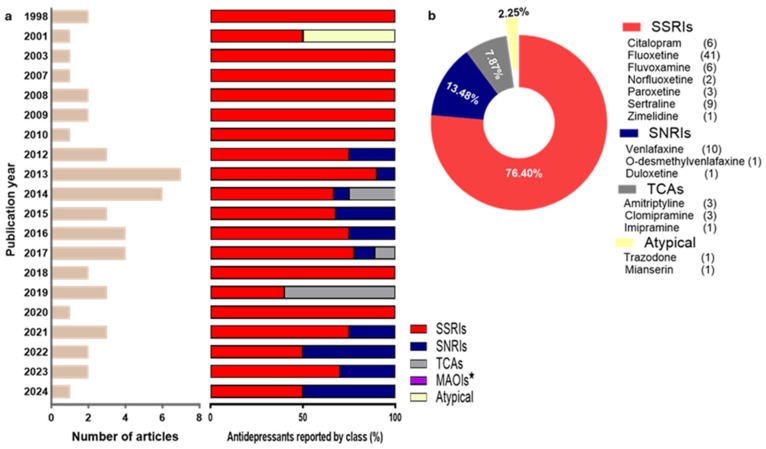
Articles included in the study and reported antidepressants: (**a**) articles included in the study by year of publication and percentage antidepressants reported by class; (**b**) total number of individual antidepressants (and percentage contribution by class) reported in all included studies. * Monoamine oxidase inhibitors (MAOIs) were not reported.

**Figure 3 toxics-13-01043-f003:**
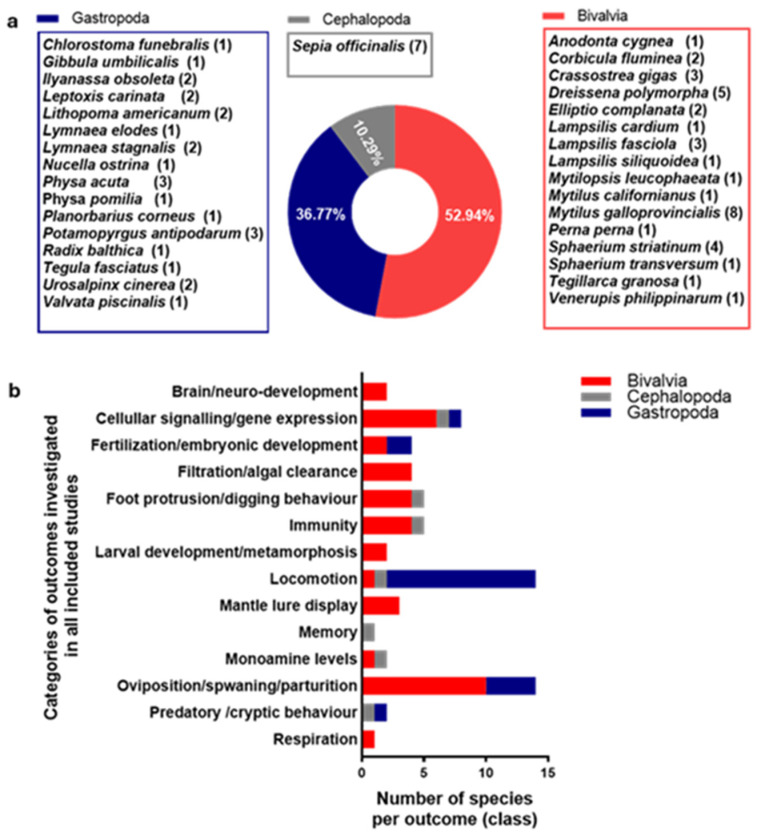
Test species and categories of measured outcomes: (**a**) test species reported in all included studies with the number of reports per species in parenthesis; (**b**) categories of measured outcomes per the number of species in which they were reported by class (bivalvia, cephalopoda and gastropoda).

**Figure 4 toxics-13-01043-f004:**
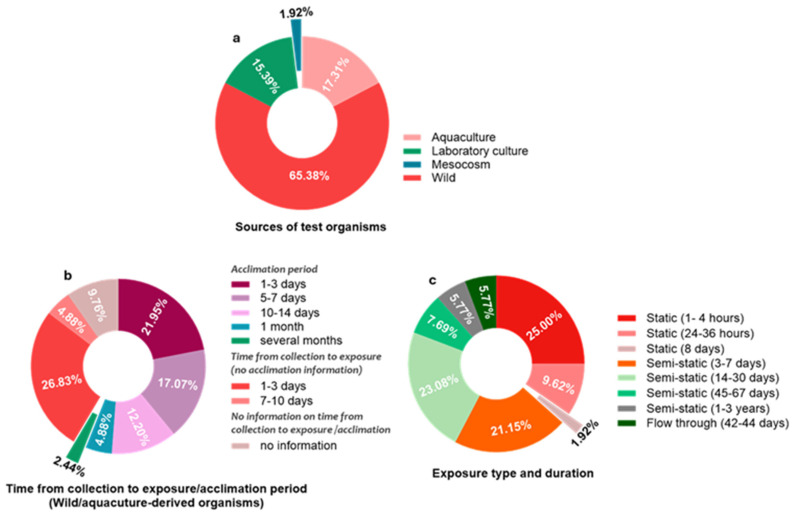
Sources of test species, acclimation and duration of exposure: (**a**) percentage of test species collected from farms (aquaculture), laboratory cultures, mesocosm and from the wild; (**b**) duration of acclimation for wild-derived test species; (**c**) duration of exposures for all included studies.

**Figure 5 toxics-13-01043-f005:**
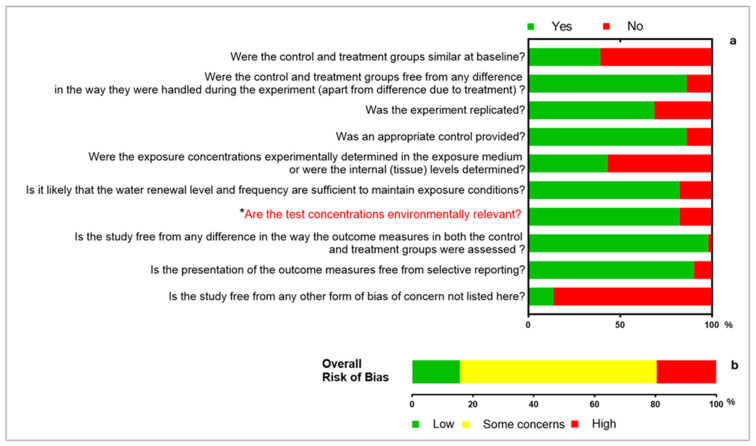
Study quality: (**a**) internal validity of all included studies with 10 signaling questions spanning several selection, performance and reporting domains; (**b**) overall study risk of bias. * Of the 42 studies (82.35%) that used environmentally relevant concentrations, 11 (21.57%) only used a single exposure concentration level, 9 (17.65%) with several exposure concentrations only had one environmentally relevant concentration level, while 5 additional studies (9.80%) with multiple bioassays only had environmentally relevant concentrations in one of the assays.

**Figure 6 toxics-13-01043-f006:**
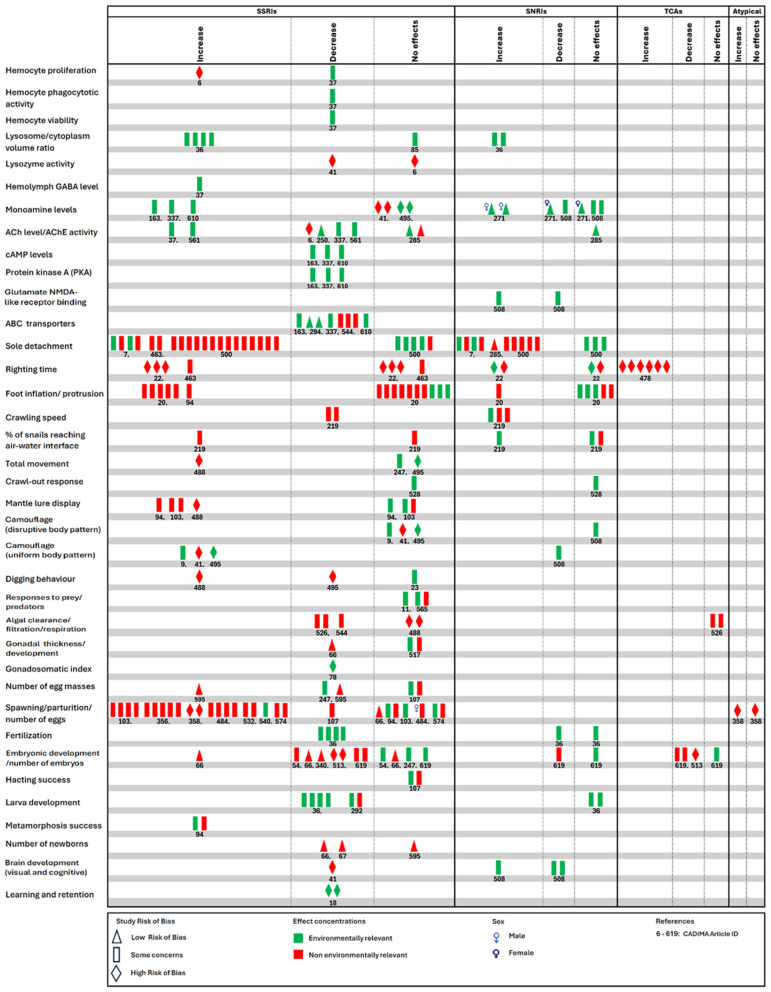
A harvest plot synthesis of study outcomes for all included studies. The synthesis-integrated study outcomes and study risk of bias with effect concentrations (environmentally relevant or higher). For each outcome level (row) per study (indicated by CADIMA article ID number), multiple outcome symbols represent different antidepressants (for the class indicated) and/or different molluscan species. The described effects (an increase or decrease relative to the control) are statistically significant. GABA: γ-aminobutyric acid; Ach: acetylcholine; AChE: acetylcholinesterase; cAMP: 3’,5’-cyclic adenosine monophosphate; NMDA: N-methyl-D-aspartate receptor; ABC transporter: ATP-binding cassette (ABC) transporters; ^Cadima ID^ [Reference]: ^6^ [124]; ^7^ [134]; ^9^ [135]; ^11^ [136]; ^18^ [137]; ^20^ [138]; ^22^ [139]; ^23^ [140]; ^36^ [86]; ^37^ [125]; ^41^ [115]; ^54^ [141]; ^66^ [142]; ^67^ [143]; ^78^ [144]; ^85^ [145]; ^94^ [107]; ^103^ [146]; ^107^ [147]; ^163^ [108]; ^219^ [148]; ^247^ [149]; ^250^ [128]; ^271^ [131]; ^285^ [150]; ^292^ [151]; ^294^ [87]; ^337^ [126]; ^340^ [152]; ^356^ [153]; ^358^ [154]; ^463^ [155]; ^478^ [156]; ^484^ [157]; ^488^ [158]; ^495^ [132]; ^500^ [159]; ^508^ [130]; ^513^ [160]; ^517^ [109]; ^526^ [110]; ^528^ [111]; ^532^ [161]; ^540^ [162]; ^544^ [112]; ^561^ [129]; ^565^ [163]; ^574^ [164]; ^595^ [165]; ^610^ [127]; ^619^ [166].

**Figure 7 toxics-13-01043-f007:**
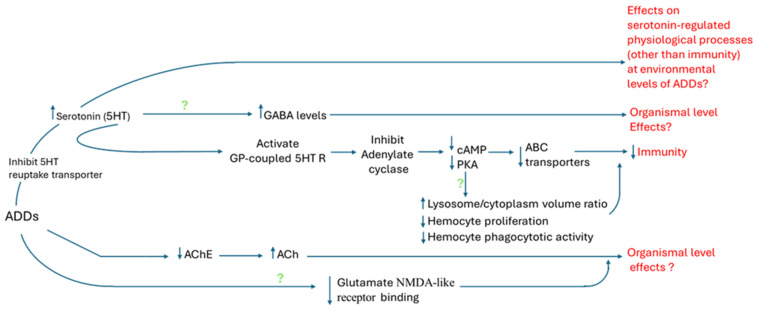
A summary of target-specific effects of environmental levels of ADDs (SSRIs and/or SNRIs) in molluscs based on current evidence. Upwards arrow: increasing effects; downwards arrow: decreasing effects; green question mark: data-deficient outcomes. 5HT: serotonin; 5HT R: serotonin receptor; GP: G. protein; GABA: γ-aminobutyric acid; cAMP: 3’,5’-cyclic adenosine monophosphate; PKA: protein kinase A; ABC transporter: ATP-binding cassette (ABC) transporters; AChE: acetylcholinesterase; Ach: acetylcholine; and NMDA: N-methyl-D-aspartate.

**Table 1 toxics-13-01043-t001:** Reported outcomes and levels of evidence across all studies for each outcome.

Reported Outcomes	Range of Evidence Levels Across All Studies for Each Outcome
Increase in hemocyte count/proliferation, foot protrusion/inflation, total movement, mantle lure display, digging behavior, number of egg masses and embryonic development; decrease in lysozyme activity, digging behavior, algal clearance/filtration, gonadal development, embryonic development, and number of newborns.	‘Generally or definitely inconclusive’
Increase in righting time	‘Generally or definitely inconclusive’ to ‘very little or no evidence’
Decrease in gonadosomatic index and retention (memory)	‘Very little or no evidence’
Increase in sole detachment, crawling speed, number of snails reaching air–water interface, spawning/parturition/number of eggs, and metamorphosis success; decrease in number of egg masses, larval and brain development.	‘Generally or definitely inconclusive’ to ‘weak to moderate evidence’
Increase in uniform body pattern (camouflage)	‘Generally or definitely inconclusive’, and ‘very little or no evidence’ to ‘weak to moderate evidence’
Increase in lysosome/cytoplasm volume ratio, hemolymph GABA level, glutamate NMDA-like receptor binding and brain development; decrease in hemocyte proliferation, hemocyte phagocytotic activity, hemocyte viability, cAMP level, protein kinase A, NMDA-like receptor binding, camouflage (uniform body pattern) and fertilization	‘Weak to moderate evidence’
Decrease in acetylcholinesterase activity and ATP-binding cassette (ABC) transporter levels.	‘Generally or definitely inconclusive’ and ‘weak to moderate evidence’ to ‘strong evidence’
Increase in monoamine levels.	‘Weak to moderate evidence’ to ‘strong evidence’

Note that the synthesized evidence is further evaluated (overall weight) by using the evidence level that has the highest number across all studies for each of the outcomes (Figure 6).

## Data Availability

Data is contained within the article and Appendix A.

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
