# Peer review of "Understanding Target-Specific Effects of Antidepressant Drug Pollution on Molluscs: A Systematic Review Report"

_toxics, 2025, doi:10.3390/toxics13121043_

Round 1

Reviewer 1 Report

Comments and Suggestions for Authors

The study aims to understand the target effects of environmental concentrations of different ADD classes in shellfish through a systematic literature review.
This appears to be the first systematic review to examine the effects of antidepressant contamination (ADD) on shellfish, an ecologically important group of animals.The authors successfully assessed the evidence for the target-specific effects of environmental ADD concentrations on shellfish, described key issues, proposed solutions, and outlined areas for future research.

The study follows the protocol of Iminwa et al., 2023.

Strengths of the study:
1. Transparency: All deviations from the protocol are clearly declared and justified.
2. Flexibility and Adaptability: The authors judiciously adjusted their methods based on real data to improve the quality and accuracy of the review. 
3. Increased Rigor: Stricter risk of bias criteria and the development of a sophisticated evidence synthesis system significantly enhance the reliability of the review's conclusions.
4. Practical Focus: Changes to the signaling questions and a focus on relevant concentrations make the review more valuable for regulatory decisions.
The authors' decision to use Cadima is a pragmatic choice of the right tool for a specific task.
The use of "gray" sources significantly increases the reliability and validity of the systematic review.

Minor comments and remarks:
A brief comment regarding "environmentally relevant concentrations": in Section 2.3, the authors set an upper limit for fluoxetine of 1 µg/L, but stipulate that for other antidepressants (venlafaxine, citalopram), they use the maximum reported levels, which may be significantly higher. This approach, although pragmatic, may lead to the inclusion of concentrations in the analysis that are more extreme outliers than typical "realistic" scenarios.

Section numbering inconsistency:
The text contains a section "1.2. Objectives of the review," but is missing a section "1.1"—1.2 follows immediately after the introduction.

Author name typo: In the "Author Contributions" section, the last line reads "project administration, E.M.I.; A.B. and E.J.R." Clearly, this refers to M.E.I. (Maurice E. Iminwa). This is a simple, yet unfortunate, typo.

Author Response

The study aims to understand the target effects of environmental concentrations of different ADD classes in shellfish through a systematic literature review.

This appears to be the first systematic review to examine the effects of antidepressant contamination (ADD) on shellfish, an ecologically important group of animals.  The authors successfully assessed the evidence for the target-specific effects of environmental ADD concentrations on shellfish, described key issues, proposed   solutions, and outlined areas for future research.

Response: Thank you for the summary of the work which confirms that the scope of the systematic review is clear.

The study follows the protocol of Iminwa et al., 2023.  Response: That is correct

Strengths of the study:

  1. Transparency: All deviations from the protocol are clearly declared and justified.
  2. Flexibility and Adaptability: The authors judiciously adjusted their methods based on real data to improve the quality and accuracy of the review.
  3. Increased Rigor: Stricter risk of bias criteria and the development of a sophisticated evidence synthesis system significantly enhance the reliability of the review's conclusions.
  4. Practical Focus: Changes to the signaling questions and a focus on relevant concentrations make the review more valuable for regulatory decisions.

The authors' decision to use Cadima is a pragmatic choice of the right tool for a specific task.

The use of "gray" sources significantly increases the reliability and validity of the systematic review.

Response: We thank the Reviewer for this summary which agrees with the strengths of the systematic review process.

Minor comments and remarks:

A brief comment regarding "environmentally relevant concentrations": in Section 2.3, the authors set an upper limit for fluoxetine of 1 µg/L, but stipulate that for other antidepressants (venlafaxine, citalopram), they use the maximum reported levels, which may be significantly higher. This approach, although pragmatic, may lead to the inclusion of concentrations in the analysis that are more extreme outliers than typical "realistic" scenarios.

Response:

We appreciate the reviewer’s thoughtful comment. Our intention in defining environmentally relevant concentrations was to balance ecological realism with the need to capture the full range of contaminant levels documented in peer-reviewed monitoring studies. For fluoxetine specifically, we applied an upper limit of 1 µg/L because this value aligns with its highest verified effluent detections, and is low compared to other antidepressants.  To guide against the introduction of unrealistic conditions, the limit was only applied to studies in which the 1 µg/L fluoxetine was analytically determined, and not nominal 1 µg/L fluoxetine (as nominal could actually be higher).

For other antidepressants such as venlafaxine and citalopram, the maximum reported surface-water levels were substantially higher and were therefore adopted to ensure that the analysis reflected the full span of real-world measurements. We acknowledge the reviewer’s concern that these maxima may represent extreme or infrequent scenarios. However, defining a threshold concentration for each individual antidepressant is inherently challenging, as environmental levels vary widely across countries, catchments, river systems, treatment processes, and even seasonally within the same location. Consequently, identifying and justifying a single representative benchmark concentration and range across studies would introduce considerable uncertainty and may not accurately reflect the diversity of environmental contexts.

Section numbering inconsistency:

The text contains a section "1.2. Objectives of the review," but is missing a section "1.1"—1.2 follows immediately after the introduction.

Response: We thank the Reviewer for spotting this error.  We have corrected the order of subheadings.

Author name typo: In the "Author Contributions" section, the last line reads "project administration, E.M.I.; A.B. and E.J.R." Clearly, this refers to M.E.I. (Maurice E. Iminwa). This is a simple, yet unfortunate, typo.

Response: We thank the Reviewer for spotting this typo which we have corrected.

Reviewer 2 Report

Comments and Suggestions for Authors

The specific effects of environmental concentrations of different classes of antidepressant drugs (ADDs) in molluscs were summarized systematically in this manuscript. This work will provide novel critical data relevant to regulatory risk decision making. More significantly, this study delineates the direction for subsequent research in the field and offers numerous constructive recommendations. It is of potential interest for publication in TOXICS. However, there are some aspects that need to be improved and a minor revision should be carried out, taking into account the specific comments.

SPECIFIC COMMENTS:

  1. The Introduction of this manuscript is excessively lengthy, which hinders readers from understanding its logical flow.
  2. The subtitle “1.2. Objectives of the review” can be found in Line 144 in. However, the subtitle of 1.1 probably be missed.
  3. The Discussion section contains several lengthy paragraphs. To enhance readability and logical coherence, it is suggested that the paragraphs be further divided and appropriate subheadings be added according to the content structure. For example, Line 652-719.
  4. The font of "glutamate NMDA-like receptor" in the Figure 7 is inconsistent with that of the other text.

Author Response

The specific effects of environmental concentrations of different classes of antidepressant drugs (ADDs) in molluscs were summarized systematically in this manuscript. This work will provide novel critical data relevant to regulatory risk decision making. More significantly, this study delineates the direction for subsequent research in the field and offers numerous constructive recommendations. It is of potential interest for publication in TOXICS. However, there are some aspects that need to be improved and a minor revision should be carried out, taking into account the specific comments.

Response: We thank the Reviewer for their comment which recognises the novelty and significance of our systematic review. 

SPECIFIC COMMENTS:

  1. The Introduction of this manuscript is excessively lengthy, which hinders readers from understanding its logical flow.

Response:
We thank the reviewer for this observation. Although the other reviewers did not raise concerns regarding the length or structure of the Introduction, we agree that improving the clarity and focus of this section will strengthen the manuscript. The original Introduction aimed to provide a comprehensive multidisciplinary context covering antidepressant pharmacology, monoamine biology, environmental occurrence, and molluscan physiology, all of which inform the rationale for this systematic review.  In response to the reviewer’s suggestion, we have carefully streamlined the Introduction by removing non-essential mechanistic detail, reducing repetition, and restructuring the material to improve the logical flow from environmental occurrence to ecological relevance and finally to the rationale for the review.

  1. The subtitle “1.2. Objectives of the review” can be found in Line 144 in. However, the subtitle of 1.1 probably be missed.

Response:  We thank the Reviewer for spotting this typo.  The numbering order has now been corrected.

  1. The Discussion section contains several lengthy paragraphs. To enhance readability and logical coherence, it is suggested that the paragraphs be further divided and appropriate subheadings be added according to the content structure. For example, Line 652-719.

Response:
We thank the reviewer for this helpful observation. In response, we have made changes to the Discussion section (including further subheadings) to improve clarity, readability, and logical flow while maintaining the scientific content and reference order.

Specifically, we have added clear and topic-appropriate subheadings throughout the Discussion to better delineate the major themes, mechanistic interpretations, and apical outcomes discussed, including addressing Lines 652-719.  We have also lightly refined sentence structure to make complex or lengthy sentences more concise and easier to follow, while keeping the meaning and nuance of the original text intact. These revisions improve the logical coherence and readability of the Discussion while preserving the full depth of analysis and scientific detail provided in the original manuscript.

  1. The font of "glutamate NMDA-like receptor" in the Figure 7 is inconsistent with that of the other text.

Author response: Thank you for spotting this, it has now been corrected.

Reviewer 3 Report

Comments and Suggestions for Authors

General comments:

This review is interesting and uses a method described by the authors in a previous publication. The presence of ADDs is the environment is of great concern for exposed biota as they are highly biologically active and the subtle effects can have major impacts on the health and functions of ecosystems. The review clearly highlights that research to date on molluscs has been inadequate to provide a solid risk assessment- the message is clearly delivered when looking at Table 1. I think that the Conclusions and Future directions section could be further developed as it is the key outcome from the review. The question that regulators and environmental managers often ask is the so what? The suggestions are mostly towards more strict experimental design and measurement. The authors should consider what is required to assess whether these stressors are actually leading to ecological changes. Are the exposed populations of molluscs surviving? If yes, are they being selected etc? There is an element of evolutionary toxicology that should be considered. As for any chemicals with low acute toxicity but highly biologically active, it would be good to consider multi-generational effects. Also is there a developmental stage when the organism is more susceptible?

Specific comments:

Line 2. It is advisable to replace “pollution” with “contaminants” in the title. The reason is that not all contaminants are pollutants. The outcome of the review definitely did not confirmed that ADDs are pollutants.

Line 13. You can probably get rid of “(the target of ADDs)” as it is repetition.

Line 16. I suggest changing “unclear” to “poorly understood”

Line 34. I find the conclusion of the Abstract to be on the emotional side. I would avoid using “…to assess and mitigate the environmental risks posed by ADDs”. I think it would be more appropriate to state that better designed studies are needed to fully characterise the hazards of ADDs at environmentally relevant concentrations and the risk they pose to exposed biota and ecosystems. The way it is written is as if the risk are already known and need addressing.

Line 36. I suggest replacing “adverse effects characterization and assessment’ with “hazard characterisation” in the keywords.

Line 47. I find “… at environmental concentrations…” misrepresentative. I suggest changing it to “…at concentrations reported/measured in the environment…” or something along this line.

Line 69. Sentences shouldn’t start with an acronym.

Line 104. Introducing the topic of molluscs so requires a new paragraph.

Line 115. “…one (estrogen receptor) does not bind to estrogen…”, isn’t it the other way around, i.e. estrogens bind to the ER??

Lines 123… This information about molluscs would fit better earlier in the intro as it would better flow and be more logical.

Line 127. Again, “pollution” here is not quite appropriate. It is a good model as it can provide good information about the effects of exposure to contaminants as they don’t move.

Line 174. I fully support that animals “…collected from the wild with little or no acclimation…” is inappropriate for toxicity testing. Maybe add here what is the recommended acclimation period as per recognised standard protocols like ASTM and OECD.

In section 3.2, it would be good to get some detail about the locations of the studies, including climate, i.e. temperate/semi-tropical etc.

In Figure 5 one key element that should be reported is the quality of the analytical chemistry support in terms of QA/QC, limits of detection etc.

Line 727-. Can you further explain what you mean by: “…for both effects and effect direction.”

Author Response

This review is interesting and uses a method described by the authors in a previous publication. The presence of ADDs is the environment is of great concern for exposed biota as they are highly biologically active and the subtle effects can have major impacts on the health and functions of ecosystems. The review clearly highlights that research to date on molluscs has been inadequate to provide a solid risk assessment- the message is clearly delivered when looking at Table 1. I think that the Conclusions and Future directions section could be further developed as it is the key outcome from the review. The question that regulators and environmental managers often ask is the so what? The suggestions are mostly towards more strict experimental design and measurement. The authors should consider what is required to assess whether these stressors are actually leading to ecological changes. Are the exposed populations of molluscs surviving? If yes, are they being selected etc? There is an element of evolutionary toxicology that should be considered. As for any chemicals with low acute toxicity but highly biologically active, it would be good to consider multi-generational effects. Also is there a developmental stage when the organism is more susceptible?

Response:
We thank the reviewer for this constructive suggestion. In response, we have substantially revised the Conclusion section to incorporate a broader ecological and regulatory perspective. Specifically, we now discuss how mechanistic and sublethal effects may (or may not) translate into population-level consequences, including survival, reproduction, and potential selection pressures under chronic environmental exposure. We have also added consideration of evolutionary toxicology, the need for multigenerational and transgenerational studies, and the importance of identifying life stages that may exhibit heightened sensitivity to antidepressants. These revisions provide a clearer context for regulators by outlining the types of data required to assess whether environmental ADD exposure is likely to lead to meaningful ecological change. The revised Conclusion appears in the updated manuscript.

Specific comments:

Line 2. It is advisable to replace “pollution” with “contaminants” in the title. The reason is that not all contaminants are pollutants. The outcome of the review definitely did not confirmed that ADDs are pollutants.

Response: We thank the reviewer for this constructive suggestion.  However, after careful consideration we have not changed the title in order to ensure consistency with the previously reported systematic review protocol.  Although your point has merit from a regulatory perspective, we cannot conclude that ADDs are simply contaminants and not pollutants either based on the evidence for some effects at environmental levels (e.g. immunity) and the fact that mixture effects have not been considered.

Line 13. You can probably get rid of “(the target of ADDs)” as it is repetition.

Response: We thank the reviewer for this constructive suggestion.  We have removed the text.

Line 16. I suggest changing “unclear” to “poorly understood”

Response: We thank the reviewer for this constructive suggestion.  We have changed the text.

Line 34. I find the conclusion of the Abstract to be on the emotional side. I would avoid using “…to assess and mitigate the environmental risks posed by ADDs”. I think it would be more appropriate to state that better designed studies are needed to fully characterise the hazards of ADDs at environmentally relevant concentrations and the risk they pose to exposed biota and ecosystems. The way it is written is as if the risk are already known and need addressing.

Response: We thank the reviewer for this their suggestion.  We have changed the abstract.

Line 36. I suggest replacing “adverse effects characterization and assessment’ with “hazard characterisation” in the keywords.

Response: We thank the reviewer for this constructive suggestion.  We have changed the keywords.

Line 47. I find “… at environmental concentrations…” misrepresentative. I suggest changing it to “…at concentrations reported/measured in the environment…” or something along this line.

Response: We thank the reviewer for this constructive suggestion.  We have made substantive changes to this section (at the request that it is shortened and made more concise by another reviewer), and have incorporated this suggestion where appropriate into the revised text.  We think these terms can be used interchangeably without any misconstruction. We have, however, tried to as much as possible to replace it,  but without the need for an exhaustive coverage as “at environmental concentrations” is extensively used throughout the text.  Also, as the use of antidepressant drugs is still increasing, we have also used the phrase “at current environmental concentrations” as this reflects that any perceived hazards could change depending on whether systematic increases in ADD consumption are reflected in systematic increases in environmental concentrations.

Line 69. Sentences shouldn’t start with an acronym.

Response: We thank the reviewer for their comment.  Our understanding is that sentences can start with an acronym that has already been defined.

Line 104. Introducing the topic of molluscs so requires a new paragraph.

Response: We have made substantive changes to this section (at the request of another reviewer that it is shortened and made more concise by another reviewer), and hope the new format is clearer.

Line 115. “…one (estrogen receptor) does not bind to estrogen…”, isn’t it the other way around, i.e. estrogens bind to the ER??

Response: We thank the reviewer for their comment. We have adjusted the text.

Lines 123… This information about molluscs would fit better earlier in the intro as it would better flow and be more logical.

Response: We have made substantive changes to this section (at the request of another reviewer that it is shortened and made more concise by another reviewer), and hope the new format is clearer.

Line 127. Again, “pollution” here is not quite appropriate. It is a good model as it can provide good information about the effects of exposure to contaminants as they don’t move.

Response: We thank the reviewer for their comment.  We consider that as a general statement, the use of the word “pollution” in this context is acceptable, as many chemicals released into the aquatic environment are hazardous.

Line 174. I fully support that animals “…collected from the wild with little or no acclimation…” is inappropriate for toxicity testing. Maybe add here what is the recommended acclimation period as per recognised standard protocols like ASTM and OECD.

Response: We thank the reviewer for their comment.  Given the scope of the review, we would prefer to keep the main emphasis on ADDs, and there only a few protocols that explicitly mention an acclimation period, whereas most do not give a fixed time, but explain that animals should be “adapted to laboratory conditions”.  We also agree with the Reviewer that this is a critical issue that deserves more consideration.  The OECD acclimation recommendation for molluscs toxicity testing has been added to the text as follows: “Standard regulatory tests with molluscs (i.e. published by the Organisation for Economic Co-operation and Development (OECD)) primarily recommend using animals which have been bred and routinely cultured under laboratory conditions. When wild-derived molluscs are needed, the OECD recommends the use of F1 offspring raised under controlled laboratory conditions, rather than wild collected F0 animals [121,122].”

In section 3.2, it would be good to get some detail about the locations of the studies, including climate, i.e. temperate/semi-tropical etc.

Response: We thank the reviewer for their comment.  The review focused on laboratory-based exposure studies – for test organisms exposed only under laboratory or, similarly controlled, conditions.  Identifying the site of animals sourced from the wild would add further complexity to the narrative which might confuse readers.  The location and temperature for all studies has not be disclosed so the purpose and value-add of this exercise would be questionable.  Climate and location was also not included as part of the original protocol.

In Figure 5 one key element that should be reported is the quality of the analytical chemistry support in terms of QA/QC, limits of detection etc.

Response:
We thank the reviewer for highlighting the importance of analytical quality assurance/quality control (QA/QC) and limits of detection in studies assessing chemical exposures. We fully agree that these elements are critical for confidence in exposure characterisation.  Quality control/quality assurance is a range of different parameters depending on the type and the intended use of analytical assays, and were not defined as components of the signalling questions in our review protocol. Extracted data on exposure and exposure determination are, however, provided as additional data in supplementary files.  We have mentioned in the conclusion section that there is a large reliance on nominal exposures and highlight the need for improved reporting of analytical QA/QC, detection limits, and exposure verification in future studies. This refinement will help regulators and researchers interpret the current evidence base with appropriate caution.

Line 727-. Can you further explain what you mean by: “…for both effects and effect direction.”

Response: We thank the reviewer for their query.  We have clarified the wording in the sentence to make it clearer.